# TransFourier: FFT Is All You Need

## Abstract

The scalability of Large Language Models (LLMs) to handle extremely long sequences is hindered by two foundational challenges: the quadratic computational cost of self-attention and the generalization limitations of positional encodings when extrapolating to contexts far beyond their training regime. These factors create bottlenecks for both the efficiency and the effective context window of current models. This paper introduces TransFourier, a novel architecture designed to address these challenges. TransFourier completely replaces the masked self-attention module with a parameter-efficient, $O(L \log L)$ Multi-Head Fourier (MHF) module. Our core contributions are threefold: (1) We propose a model that leverages the Fast Fourier Transform (FFT) for sequence information mixing, inherently addressing the aforementioned computational and generalization bottlenecks of attention. (2) We introduce a novel frequency-domain causal masking technique, which elegantly enforces autoregressive capabilities through asymmetric padding and truncation, overcoming a critical barrier that has historically limited Fourier-based models in generative tasks. (3) Our design is built entirely on highly-optimized, standard deep learning operators (e.g., FFT and convolution), obviating the need for hardware-specific custom CUDA kernels, unlike architectures such as Mamba, thus ensuring broad accessibility and portability. Evaluations on established academic benchmarks show that TransFourier is competitive with mature Transformer and State Space Model (SSM) baselines of comparable size. Given its strong scaling and architectural simplicity, TransFourier presents a compelling and practical pathway toward developing the next generation of efficient long-sequence models. The code is available in the supplementary materials.

## 1 Introduction

The Transformer architecture (Vaswani et al., 2017) has become a foundational model for a wide range of sequence modeling tasks. However, the remarkable success of this paradigm is built upon self-attention mechanism that presents two fundamental limitations. The first is computational complexity: the number of operations in self-attention scales quadratically with the input sequence length $L$, i.e., $O(L^2)$. This scaling creates a significant computational bottleneck that makes processing long sequences prohibitively expensive. The second limitation is its inherent permutation equivariance; self-attention is insensitive to the order of tokens, which necessitates the introduction of external positional encodings to provide the model with sequential information. These encodings, whether absolute or relative, tend to tether the model's performance to the context lengths seen during training. When extrapolating to much longer contexts, this often leads to performance instability and degradation, a phenomenon that effectively imposes a context window limit on the model.

One major line of research, the *Enhancement Paradigm*, aims to mitigate these issues by modifying the attention framework. To address the computational cost, sparse attention methods like Longformer (Beltagy et al., 2020) and BigBird (Zaheer et al., 2020) introduce fixed, sparse connectivity patterns. While effective, these methods rely on heuristics and risk information loss. Concurrently, sophisticated relative positional encodings like RoPE (Su et al., 2024), ALiBi (Press et al., 2022), and YaRN (Peng et al., 2024), which encode relative distances or apply distance-aware biases, have dramatically improved length extrapolation. However, the growing complexity of these techniques shows they are sophisticated compensations for a mechanism lacking a built-in sequence concept, motivating the search for a more fundamental alternative.

A more radical approach, the *Replacement Paradigm*, replaces the attention mechanism with more efficient alternatives. State Space Models (SSMs), particularly the recent Mamba architecture (Gu & Dao, 2023; Dao & Gu, 2024), have emerged as a powerful contender. Mamba achieves linear-time complexity and state-of-the-art performance, but its efficiency heavily relies on hardware-specific custom CUDA kernels, hindering portability, modification, and broad adoption. Another prominent direction within this paradigm leverages the Fourier Transform, offering an appealing $O(L \log L)$ complexity for global token mixing (Lee-Thorp et al., 2022). However, these Fourier-based models have historically struggled with generative tasks. Their primary flaw lies in the difficulty of enforcing causality—a strict requirement for autoregressive decoding—within the frequency domain, a challenge that has relegated them to encoder-only or non-autoregressive applications.

In this work, we argue that a solution can be found that is both algorithmically elegant and hardware-agnostic. We introduce TransFourier, a novel architecture that directly confronts the foundational issues of attention. By replacing the attention module with our Multi-Head Fourier module, Trans-Fourier inherently resolves both the quadratic complexity and the need for positional encodings. Furthermore, by developing a novel causal masking technique that operates purely in the frequency domain, we overcome the critical causality barrier that has hindered previous Fourier-based models. Our contributions are as follows:

- **A Novel Autoregressive Fourier Module:** We propose the Multi-Head Fourier (MHF) module, which mixes token information via a gated element-wise product in the frequency domain, directly replacing the attention layer in a Transformer.
- **Frequency-Domain Causal Masking:** We introduce a technique that enforces causality entirely in the frequency domain using asymmetric padding and truncation. This addresses a key obstacle that has hindered the application of Fourier models to generative tasks.
- **A Position-Encoding-Free Architecture:** Our model obviates the need for explicit positional encoding because the Fourier Transform inherently captures position through its sinusoidal basis functions. This design is theoretically advantageous for length extrapolation.
- **Solid Validation:** On various public datasets, we demonstrate that TransFourier is a highly competitive architecture, outperforming standard Transformers and posing a strong challenge to state-of-the-art SSMs, all while offering greater implementational simplicity.

## 2 RELATED WORK

To overcome the foundational limitations of the standard Transformer, several active lines of research have emerged. These can be broadly categorized by their core strategy: improving existing components like attention and positional encodings, or replacing them entirely with new architectural primitives. In this section, we review the most prominent directions to situate our work.

### 2.1 SPARSE ATTENTION

To address the prohibitive $O(L^2)$ complexity of self-attention on long sequences, a significant body of work has explored sparse attention patterns. Seminal models like Longformer (Beltagy et al., 2020) introduced handcrafted sparsity by combining local windowed attention with a small number of global tokens. Other approaches have explored random patterns as part of a hybrid strategy (Zaheer et al., 2020), or employed adaptive, content-based sparsity through techniques like locality-sensitive hashing in Reformer (Nikita et al., 2020) and learned routing in Routing Transformer (Aurko et al., 2021). While these methods successfully lower the computational burden, they do so at a cost. By design, they break the fully-connected graph of standard attention and often introduce new, sensitive hyperparameters, potentially impeding model performance.

### 2.2 RELATIVE POSITIONAL ENCODINGS

To address the poor length generalization caused by positional encodings, a family of sophisticated relative positional encodings (RPEs) has been developed. ALiBi (Press et al., 2022) introduces a simple, static bias to attention scores based on token distance, demonstrating remarkable extrapolation capabilities. RoPE (Su et al., 2024) applies rotations to queries and keys in a way that

makes attention scores sensitive only to relative positions. Subsequent work, such as YaRN (Peng et al., 2024), has further refined these rotational methods to achieve even more robust extrapolation. Ultimately, while highly effective, these methods function as sophisticated patches to address the symptoms of attention's permutation equivariance, rather than providing a root solution.

## 2.3 STATE SPACE MODELS (SSMs)

A more radical approach is to replace the attention mechanism altogether with a new sequence mixing primitive. SSMs have recently emerged as a powerful and efficient replacement for attention. Foundational work like the Structured State Space model (S4) (Gu et al., 2022b) and Diagonal State Space model (S4D) (Gu et al., 2022a) first demonstrated their potential for modeling long-range dependencies by leveraging a formulation that could be efficiently computed in parallel as a global convolution. Crucially, this approach was purely algorithmic and hardware-agnostic. However, the time-invariant nature of S4's state matrices limited its ability to perform content-aware reasoning and select relevant information.

This limitation was addressed by successors like Mamba (Gu & Dao, 2023) and Mamba-2 (Dao & Gu, 2024), which introduced selective, input-dependent state updates. This innovation dramatically improved performance but came at a significant cost: the loss of the parallel convolutional form. To maintain efficiency, Mamba relies on a hardware-aware sequential scan algorithm. This dependency is the primary drawback of modern selective SSMs. Their impressive efficiency is not purely algorithmic but a result of tight hardware-software co-design, requiring custom CUDA and Triton kernels optimized for specific NVIDIA GPUs. This creates a significant barrier to entry, hindering portability and complicating algorithmic modifications, in sharp contrast to the universal compatibility of standard library operators.

## 2.4 FOURIER AND SPECTRAL METHODS

Another major line of research leverages the Fourier Transform and other spectral methods, motivated by their ability to achieve global token mixing with an efficient $O(L \log L)$ complexity. These works can be divided into two main categories: those that use spectral methods to enhance existing architectures, and those that use them to completely replace attention.

### 2.4.1 SPECTRAL METHODS AS ENHANCEMENTS

This category of models integrates Fourier-based operators into an existing architecture, typically a Transformer, to improve efficiency or performance. Fourier Transformer (He et al., 2023) and FwNet-ECA (Mian et al., 2025) insert a Fourier block after attention layers to compress the sequence representation, thereby reducing the computational load on subsequent layers. Others use spectral mixing as a parallel path; for example, Vim-F (Zhang et al., 2024) augments a Mamba block with a parallel Fourier filtering module for vision tasks.

Some works use spectral methods to assist attention in other ways. FSAT (Zhuang et al., 2022) uses a Fourier-based convolution to efficiently predict a sparse attention mask. FourierNAT (Kiruluta et al., 2025) employs a Fourier-mixing block in a decoder, but is limited to non-autoregressive generation, as its global FFT operation violates causality. Finally, some works like FAN (Dong et al., 2024) explore replacing activation functions in MLPs with trigonometric functions, addressing the feed-forward network rather than the attention bottleneck. While these methods demonstrate the utility of spectral biases, they do not fundamentally alter or remove the core attention mechanism.

### 2.4.2 SPECTRAL METHODS AS REPLACEMENTS

More ambitious approaches use Fourier-based modules to completely replace self-attention. The seminal work in this area, FNet (Lee-Thorp et al., 2022), showed that a parameter-free 2D Fourier Transform could substitute for attention in a BERT-style encoder, retaining competitive performance at a much higher speed. This concept was extended in vision models like GF-Net (Rao et al., 2021), which replaced attention with a learnable global filter in the frequency domain. This idea of frequency-domain filtering was inspired by the Fourier Neural Operator (FNO) (Li et al., 2021), which applied it to solving differential equations. AFNO (Guibas et al., 2022) later adapted this operator to be more efficient for high-resolution image tasks.

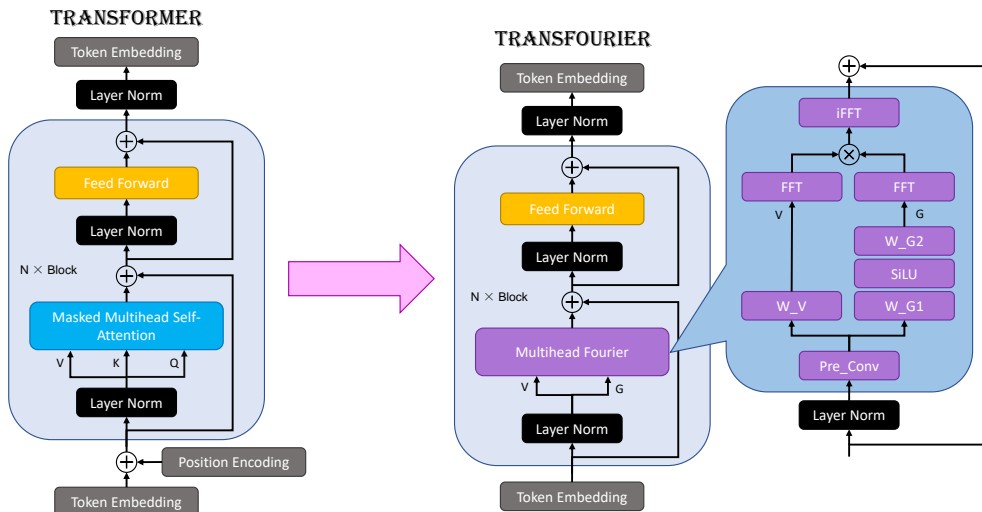

Figure 1: Comparison of a standard Transformer model (left) and our proposed TransFourier model (right). The core modification is the replacement of Masked Multi-Head Attention with our Multi-Head Fourier (MHF) module and the removal of positional encodings.

While powerful, these replacement methods have faced two major challenges. First, most, including FNet, GF-Net, AFNO, and DCT-Former (Scribano et al., 2023) (which approximates attention via DCT), are inherently non-causal and thus designed for encoder-only or vision applications. They cannot be directly applied to autoregressive sequence generation. Second, the few attempts at autoregressive replacements, such as SPECTRE (Fein-Ashley et al., 2025), have resorted to processing inputs in fixed-length sliding windows. This design reintroduces locality constraints, can fragment long-range dependencies, and may cause signal distortion when padding shorter sequences. Therefore, a truly causal, efficient, and hardware-agnostic Fourier-based replacement for attention remains an open challenge—one that our work aims to address.

## 3 METHODOLOGY

### 3.1 OVERALL STRUCTURE

The design of TransFourier prioritizes simplicity and efficiency. We take a standard decoder-only Transformer (e.g., GPT-2) as a blueprint and apply minimal modifications. As illustrated in Figure 1, a TransFourier model is structurally very similar to a Transformer model, with two key differences:

1. We replace the Masked Multi-Head Self-Attention module with our proposed **Multi-Head Fourier (MHF)** module.
2. We completely remove the **Positional Encoding** module.

This design allows TransFourier to easily leverage mature components from the existing Transformer ecosystem, such as the feed-forward network (FFN), layer normalization, and residual connections.

### 3.2 MULTI-HEAD FOURIER (MHF) MODULE

The MHF module is the heart of TransFourier, responsible for performing global, causal information mixing in $O(L \log L)$ complexity. Given an input $x \in \mathbb{R}^{B \times L \times D}$, where $B$ is the batch size, $L$ is the sequence length, and $D$ is the model dimension, the forward pass proceeds as follows:

**Step 1: Injecting Local Inductive Bias with Causal Convolution.** We first pass the input $x$ through a lightweight, depthwise causal 1D convolution. This convolution uses a small kernel (e.g., 3) and left-sided padding (causal padding, e.g., 2) to ensure that the output at each timestep depends only on current and past inputs. This step explicitly captures basic syntactic patterns (like n-grams), compensating for the removal of explicit positional encodings and allowing the subsequent global Fourier Transform to focus on longer-range semantic dependencies.

**Step 2: Preparing Gated Signals.** After layer normalization, we project the input signal $x_{norm}$ into two parallel streams: a content stream $x_v$, which carries the semantic information to be mixed, and a stream that will form our gate, $x_g$. The content stream $x_v$ is generated by passing $x_{norm}$ through a single linear layer. To form the gate $x_g$, $x_{norm}$ is passed through a linear layer and a SiLU activation, followed crucially by a point-wise (kernel_size=1) 1D convolution. With its `groups` parameter set to the number of heads ($n_{head}$), it performs intra-head channel mixing. This enables the dimensions within each head to collectively learn a shared representation, a critical step before they are processed independently by the subsequent dimension-wise Fourier Transform.

**Step 3: Causal Mixing in the Frequency Domain.** This is the core innovation of our architecture. By the Convolution Theorem, convolution in the time domain is equivalent to element-wise multiplication in the frequency domain. To implement a causal convolution efficiently, we employ a classic signal processing technique: we pad the input sequences ($x_v$ and $x_g$) to twice their length ($N = 2L$) before applying the Fast Fourier Transform (FFT) to get $V_{fft}$ and $G_{fft}$, respectively. In this expanded frequency space, we perform an element-wise product of $V_{fft}$ and $G_{fft}$ to get $X_{fft}$. Here, $G_{fft}$ acts as a dynamic filter, adaptively amplifying or suppressing specific frequency components in $V_{fft}$ based on the input content.

**Step 4: Causal Reconstruction and Projection.** We transform the multiplied spectrum $X_{fft}$ back to the time domain using the inverse FFT (iFFT). The crucial step is to truncate the resulting sequence of length $2L$ back to the original length $L$. This complete "pad-FFT-multiply-iFFT-truncate" pipeline is mathematically equivalent to performing a causal convolution between $x_v$ and $x_g$ in the time domain. This guarantees that the output at time $t$ depends strictly on inputs from $1, \ldots, t$, perfectly satisfying the requirements of an autoregressive model. Finally, a linear projection layer integrates the mixed information and outputs it to the next layer.

To provide a concrete overview of the data flow within the MHF module, we present its forward pass as pseudocode in Algorithm 1.

## 3.3 ARCHITECTURAL PROPERTIES

**Computational Complexity:** The main computational cost of the MHF module comes from FFT and iFFT, each with a complexity of $O(L \log L)$, a significant improvement over attention's $O(L^2)$.

**Implicit Positional Information:** The basis functions of the Fourier Transform (sine and cosine waves) inherently contain ordered frequency and phase information. Hence, the model can implicitly perceive token positions without external positional encodings.

**Infinite Context Capability:** Since there are no parameters in the model tied to a maximum length (such as a fixed-size positional encoding table or rotation angles calibrated for a specific length), TransFourier can theoretically process sequences of any length. In practice, its context window is limited only by hardware memory.

## 4 CONCEPTUAL ANALYSIS

In this section, we delve deeper into the architectural principles of TransFourier, providing a first-principles comparison to attention, analyzing how our design overcomes the challenge in prior spectral models, and showing how our method of data-dependent mixing fits in with the SSMs.

## 4.1 A FIRST-PRINCIPLES VIEW: ATTENTION AND FFT AS WEIGHTED SUMS

From first principles, both self-attention and the Fourier Transform can be understood as mechanisms for token mixing via a weighted sum. This fundamental similarity provides the theoretical grounding for why one can replace the other.

In a single head of self-attention, the updated representation for the $t$-th token, $r_t$, is a weighted sum of the value vectors $v_j$ from all N tokens in the context:

$$r_t = \sum_{j=0}^{N-1} \alpha_{tj} v_j \tag{1}$$

Here, the weights $\alpha_{tj}$ are computed dynamically based on the query of the $t$-th token and the key of the $j$-th token ($\alpha_{tj} = \text{softmax}(q_t \cdot k_j^T / \sqrt{d_k})$). These weights are data-dependent, forming a dense $N \times N$ attention matrix.

To understand why the Fourier Transform can serve as an alternative, it is instructive to deconstruct its definition from the same perspective. Let us consider the 1D Discrete Fourier Transform (DFT). Given a sequence of input vectors $v_0, v_1, \ldots, v_{N-1}$ (where we analyze each dimension independently), the DFT produces a sequence of output vectors $r_0, r_1, \ldots, r_{N-1}$. According to the definition of Fourier Transform, the standard formula for the $t$-th output vector is:

$$r_t = \sum_{j=0}^{N-1} v_j \cdot e^{-i \frac{2\pi}{N} tj} = \sum_{j=0}^{N-1} w_{tj} v_j \tag{2}$$

The Fourier Transform is analogous to self-attention as they share the same fundamental operation: a complete mixing of token information via a weighted sum. In the Fourier Transform, the weights are the complex exponentials, $w_{tj} = e^{-i \frac{2\pi}{N} tj}$. This equivalence, underscored by the success of Transformers, establishes the Fourier Transform as a powerful and highly efficient baseline for the token mixing operation.

### 4.2 THE CAUSALITY DILEMMA IN AUTOREGRESSIVE SPECTRAL MODELS

The primary reason why previous attempts to replace attention with FFTs have struggled in generative models lies in the challenge of enforcing causality efficiently. An autoregressive model must ensure that the prediction for token $t$ is generated using only information from tokens $0, \ldots, t$. Self-attention elegantly solves this with a causal mask. The key is that the attention weights $\alpha_{tj}$ are explicitly computed as an intermediate matrix **before** the final weighted sum is performed. This creates a crucial window of opportunity to intervene: the mask forcibly sets all weights where $j > t$ to zero, ensuring future tokens do not contribute to the output for token $t$. This is all performed in a single, parallel forward pass, allowing for highly efficient training.

A naive application of the FFT does not afford this luxury. The Fast Fourier Transform is a highly optimized algorithm that computes the full weighted sum (Eqn. 2) directly from the input sequence. **There is no intermediate step where a modifiable matrix of explicit weights ($w_{tj}$) is formed. One cannot apply a mask to the Fourier basis functions 'midway' through the computation; the algorithm yields the final sum in what is essentially a single, atomic operation.**

Since the weights themselves cannot be masked, the only way to enforce causality is to manipulate the input sequence. To generate the correct output for token $t$, one must feed the FFT algorithm only the causal portion of the sequence ($v_0, \ldots, v_t$) while hide the portion after token $t$ ($v_{t+1}, \ldots, v_{N-1}$). This leads to a significant computational inefficiency during training. To get the outputs for all $N$ tokens in a sequence, one would need to perform $N$ separate FFT computations of increasing length. The total complexity for a single training step would approach $\text{O}(N^2 \log N)$, which is significantly less efficient than the single pass of self-attention $\text{O}(N^2)$. This computational dilemma is the fundamental reason why most prior spectral models were confined to non-causal encoders or processed inputs in fixed-length, non-causal chunks.

TransFourier resolves this dilemma through its equivalence to causal convolution. The "pad-FFT-multiply-iFFT-truncate" procedure is a mathematically exact and parallelizable method for computing a causal output. This allows our model, like self-attention, to process the entire sequence in a single forward pass during training while guaranteeing that causality is strictly maintained.

### 4.3 ACHIEVING DATA-DEPENDENT MIXING WITH ALGORITHMIC EFFICIENCY

A key factor in the performance of modern sequence models is their ability to perform data-dependent or content-aware reasoning. The evolution from S4 to Mamba provides a clear illus-

tration of this principle. S4, with its time-invariant state matrices, was algorithmically efficient and could be computed as a parallel convolution. However, its data-independent nature limited its expressive power. Mamba's core innovation was to introduce selective, input-dependent state updates, dramatically improving performance. This improvement, however, came at the cost of breaking the parallel convolutional structure, necessitating a hardware-aware sequential scan algorithm that relies on custom, low-level optimizations.

TransFourier offers an alternative, more direct path to achieving data-dependent mixing while preserving algorithmic efficiency. In our architecture, the effective "convolutional kernel" is the gate stream $x_g$, which is generated dynamically from the input $x$ itself via a small neural network (Linear $\rightarrow$ SiLU $\rightarrow$ Conv1d). This makes the mixing operation fully data-dependent.

Crucially, because this interaction is formulated as a multiplication in the frequency domain, it remains equivalent to a convolution in the time domain. This formulation preserves the globally parallel computational structure that is inherent to convolutions and FFTs. Consequently, TransFourier benefits from the expressive power of data-dependent mixing, much like Mamba, but without sacrificing the hardware independence and universal parallelism of standard library operators. It achieves content-aware reasoning through a purely algorithmic and elegant mechanism.

## 5 EXPERIMENTS

To validate the effectiveness and scalability of TransFourier, we conduct a series of pre-training experiments and evaluate our models against strong, established baselines. Our experiments are designed to assess three key aspects: (1) performance on standard common-sense reasoning benchmarks compared to Transformer and SSM architectures, (2) scaling properties as model size increases, and (3) qualitative and quantitative performance on long-context tasks.

### 5.1 EXPERIMENTAL SETUP

**Dataset:** We pre-train all models from scratch on the **FineWeb-10B** dataset (Penedo et al., 2024), a high-quality, 10-billion-token corpus of English web text. This dataset is a filtered subset of Common Crawl and has become a standard resource for training foundation models of this scale.

**Baselines:** We compare TransFourier against two strong architectural paradigms:

- **Transformer:** We use a standard GPT-2 model (Radford et al., 2019), implemented via the `transformers` library. This represents the canonical attention-based architecture.
- **State Space Models (SSMs):** We use Mamba (Gu & Dao, 2023) and Mamba-2 (Dao & Gu, 2024) as our state-of-the-art SSM baselines, using the official `mamba_ssm` library. These models represent the cutting edge in efficient, non-attentional sequence modeling.

**Configurations:** To analyze scaling properties, we train all four models (TransFourier, GPT-2, Mamba, Mamba-2) at three different sizes, which we term "Mini", "Small", and "Medium". The configurations are chosen to align with the models described in the GPT-3 paper (Brown et al., 2020), ensuring a fair comparison of parameter efficiency. Specific hyperparameters are detailed in Table 1. All models were trained on a node of 8 NVIDIA GeForce RTX 3090 GPUs (24GB).

Table 1: Hyperparameter configurations for the different model sizes used in our experiments.

| Model Size | $d_{\mathbf{model}}$ | $n_{\mathbf{layer}}$ | $n_{\mathbf{head}}$ | $d_{\mathbf{head}}$ |
|---|---|---|---|---|
| Mini | 512 | 12 | 8 | 64 |
| Small | 768 | 12 | 12 | 64 |
| Medium | 1024 | 24 | 16 | 64 |

**Training Details:** We trained all models for 10B tokens with a context length of 1024. We employed the AdamW optimizer with $\beta_1 = 0.9$, $\beta_2 = 0.95$, and a weight decay of 0.1. The training utilized a global batch size of 0.5M tokens. The learning rate followed a cosine schedule, peaking at $9 \times 10^{-4}$ after a linear warmup over the first 3.75% of training steps. We also applied gradient clipping with a norm of 1.0. The `gpt2` tokenizer was used throughout all training and evaluation stages.

## 5.2 EFFECTIVENESS AND SCALING ANALYSIS: LM-EVALUATION-HARNESS

We use the `lm-evaluation-harness` framework for standardized and reproducible evaluation. We report performance on a suite of popular common-sense reasoning benchmarks: **Hellaswag** (Zellers et al., 2019), **ARC-Easy** and **ARC-Challenge** (Clark et al., 2018), and **Winogrande** (Sakaguchi et al., 2020).

Table 2 presents the main results across all models and benchmarks. Our proposed TransFourier architecture demonstrates highly competitive performance. At all scales, TransFourier models consistently outperform the GPT-2 baselines of equivalent size, highlighting the effectiveness of the MHF module as a replacement for self-attention. Furthermore, TransFourier's performance is on par with, and in some cases exceeds, the strong Mamba and Mamba-2 baselines. This indicates that our hardware-agnostic, FFT-based approach is a compelling alternative to specialized SSMs.

Table 2: Main performance comparison on common-sense reasoning benchmarks. We report accuracy (%) for all tasks. Bold numbers indicate the best performance within each parameter class.

| Size | Model | Hellaswag | ARC-e | ARC-c | Winogrande | Average |
|------|-------|-----------|-------|-------|------------|---------|
| Mini | GPT-2 | 28.44 | 43.27 | 23.89 | 48.22 | 35.96 |
| | Mamba | **30.00** | 43.39 | **25.09** | 50.20 | **37.17** |
| | Mamba-2 | 28.96 | 42.59 | 24.83 | **50.28** | 36.67 |
| | TransFourier (Ours) | 29.27 | **43.48** | 23.12 | 49.49 | 36.34 |
| Small | GPT-2 | 30.78 | **47.01** | 24.74 | 49.96 | 38.12 |
| | Mamba | **32.19** | 46.72 | 25.34 | **51.07** | **38.83** |
| | Mamba-2 | 31.04 | 45.88 | 25.43 | 50.91 | 38.32 |
| | TransFourier (Ours) | 31.34 | 46.97 | **25.51** | 50.83 | 38.66 |
| Medium | GPT-2 | 35.90 | 51.81 | 25.26 | 51.30 | 41.07 |
| | Mamba | **37.29** | 49.96 | **26.96** | 51.54 | 41.44 |
| | Mamba-2 | 36.82 | 50.29 | 26.19 | 51.93 | 41.31 |
| | TransFourier (Ours) | 36.05 | **51.85** | 25.77 | **52.17** | **41.46** |

As shown in Table 2, TransFourier exhibits a smooth and predictable scaling trend. The consistent increase in validation accuracy as model size increases demonstrates that our architecture is scalable and benefits from increased parameter counts, similar to Transformers. This is a crucial property, suggesting that TransFourier is a suitable candidate for training even larger models in the future.

## 5.3 LONG-CONTEXT EVALUATION: NEEDLE IN A HAYSTACK TEST

We conducted the Needle in a Haystack test (Kamradt, 2023) with the standard `NeedleHaystack` toolkit to evaluate the model's long-context retrieval capabilities, a task where our position-encoding-free architecture is theoretically advantageous. Our test setting involved models trained with a context length of 1024. We then evaluated their ability to retrieve the "needle" from "haystacks" of 1000 (interpolation) and 2000 (extrapolation) tokens, placing the needle at 10%, 50%, and 90% depth within the context.

Notably, the standard GPT-2 baseline could not complete this evaluation. Due to its fixed positional encodings, it raised an error and failed when processing sequences longer than its training length, highlighting a fundamental architectural limitation of traditional Transformers. In contrast, the models capable of extrapolation—TransFourier, Mamba, and Mamba-2—successfully processed the longer sequences. However, across all scales and test configurations, they performed poorly, achieving near-zero retrieval accuracy.

We hypothesize that this is not a failure of any specific architecture, but rather a consequence of the relatively small model scale (< 500M parameters). It is widely observed that robust long-context reasoning is an emergent capability that appears in much larger models. While our models did not demonstrate this capability, the fact that they correctly processed contexts beyond their training length (unlike GPT-2) and that state-of-the-art SSM baselines also failed the retrieval task under

identical conditions suggests that larger-scale experiments are necessary to fully unlock and validate the long-context potential of TransFourier.

### 5.4 ABLATION STUDY: THE ROLE OF THE FEED-FORWARD NETWORK

We select a SwiGLU-based feed-forward network (FFN) (Shazeer, 2020) for our architecture, as its gating mechanism is conceptually consistent with the dynamic, input-dependent filtering principle of our Multi-Head Fourier (MHF) module. To evaluate the impact of this choice, we conducted an ablation study across all three model scales: Mini, Small, and Medium. For each scale, we trained a counterpart model where the SwiGLU FFN was replaced with a standard MLP.

The results, presented in Table 3, show a consistent pattern across all scales. The SwiGLU-based FFN provides a discernible but modest performance improvement over the standard MLP. This consistent, small performance gap across different model sizes strongly suggests that the core effectiveness of our model is primarily driven by the MHF module itself. This finding highlights the robustness of the MHF module, which performs well even with a simpler FFN, thereby isolating the MHF's contribution as the main architectural innovation.

Table 3: Ablation study on the FFN in TransFourier models across different scales. We compare a standard MLP against our proposed SwiGLU-based FFN. Results are reported as accuracy (%).

| Model Configuration | Hellaswag | ARC-e | ARC-c | Winogrande | Average |
|---|---|---|---|---|---|
| TransFourier-Mini (MLP) | 28.60 | 41.54 | 24.40 | 49.33 | 35.97 |
| TransFourier-Mini (SwiGLU) | 29.27 | 43.48 | 23.12 | 49.49 | 36.34 |
| TransFourier-Small (MLP) | 31.30 | 44.74 | 24.83 | 51.93 | 38.20 |
| TransFourier-Small (SwiGLU) | 31.34 | 46.97 | 25.51 | 50.83 | 38.66 |
| TransFourier-Medium (MLP) | 35.63 | 50.00 | 26.11 | 52.25 | 41.00 |
| TransFourier-Medium (SwiGLU) | 36.05 | 51.85 | 25.77 | 52.17 | 41.46 |

## 6 LIMITATIONS AND FUTURE WORK

Our primary experiments validate the architecture's effectiveness up to a certain scale. To prove its capability to support frontier models at the trillion-parameter level, much larger-scale training experiments will be necessary. Furthermore, while the frequency domain offers some interpretability (e.g., analyzing which frequencies are amplified by the gating mechanism), it is generally less intuitive than attention maps. Developing better tools to visualize and understand information flow in the frequency domain is a promising direction for future research. Other avenues include exploring more sophisticated gating mechanisms and applying the architecture to multimodal data, where Fourier methods have a rich history.

## 7 CONCLUSION

In this paper, we addressed the core bottlenecks of computation and context length in modern Large Language Models by proposing TransFourier, a novel architecture. By replacing self-attention with a Fourier-based gated mixing module and eliminating positional encodings entirely, TransFourier reduces computational complexity from $O(L^2)$ to $O(L \log L)$ while theoretically enabling infinite context length. Our key technical innovation, a frequency-domain causal masking technique, elegantly resolves the long-standing challenge of applying spectral methods to generative tasks.

Our model design adheres to principles of simplicity and generality, relying exclusively on standard, optimized operators found in deep learning libraries, which ensures excellent portability and ease of use. Experimental results show that TransFourier exhibits robust scaling properties and achieves performance competitive with mainstream Transformer and SSM models on standard benchmarks. In summary, TransFourier is more than just another efficient model variant; it is a simple, powerful, and practical architectural foundation that opens a promising new path for the development of next-generation long-context sequence models.

REPRODUCIBILITY STATEMENT

To ensure the reproducibility of our work, we provide comprehensive details on our code, data, and experimental setup. The complete source code for the TransFourier model and baselines used in experiements (GPT-2, Mamba, Mamba-2), along with the scripts for pre-processing the FineWeb-10B training data, is included in the supplementary materials. All datasets used for training and evaluation are publicly available, with detailed descriptions provided in Section 5.1. Our evaluation was conducted using the standard toolkit on unmodified benchmark datasets shown in Section 5.2 and Section 5.3. Furthermore, all model configurations, hyperparameters, and specific training procedures are detailed in Section 5.1 and Table 1. We believe these resources offer a complete and clear basis for reproducing our findings.

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

## A  APPENDIX - ALGORITHM

To clarify the operational flow of our proposed Multi-Head Fourier (MHF) module, we present its forward pass in Algorithm 1. This algorithm details how causal convolution is integrated with frequency-domain mixing to model sequence data.

---

**Algorithm 1** The Multi-Head Fourier (MHF) Module Forward Pass

---

**Input**: Input sequence $x \in \mathbb{R}^{B \times L \times D}$
**Parameters**: All model weights $\theta$
**Output**: Output sequence $y \in \mathbb{R}^{B \times L \times D}$

1: **Inject local inductive bias with causal convolution. Left-padding (pad=2) for causality.**
2: $x \leftarrow \text{Conv1d}(x, \text{kernel\_size} = 3, \text{groups} = D)$
3: $x_{\text{norm}} \leftarrow \text{LayerNorm}(x)$
4: **Prepare content stream ($x_{\mathbf{v}}$) and gate stream ($x_{\mathbf{g}}$).**
5: $x_{\text{v}} \leftarrow \text{Linear}(x_{\text{norm}})$
6: $x_{\text{g}} \leftarrow \text{Conv1d}(\text{SiLU}(\text{Linear}(x_{\text{norm}})), \text{kernel\_size} = 1, \text{groups} = h)$
7: **Perform causal mixing in frequency domain.**
8: $N \leftarrow 2 \times L$                           *Set padded length for causal convolution via FFT*
9: $V_{\text{fft}} \leftarrow \text{RFFT}(x_{\text{v}}, \text{n} = N, \text{dim} = 1)$          *Pad and transform content stream*
10: $G_{\text{fft}} \leftarrow \text{RFFT}(x_{\text{g}}, \text{n} = N, \text{dim} = 1)$          *Pad and transform gate stream*
11: $X_{\text{fft}} \leftarrow V_{\text{fft}} \odot G_{\text{fft}}$         *Element-wise product in frequency domain*
12: $x_{\text{mixed}} \leftarrow \text{IRFFT}(X_{\text{fft}}, \text{n} = N, \text{dim} = 1)$   *Transform mixed spectrum back to sequence domain*
13: **Truncate to enforce causality and project to output**
14: $x_{\text{causal}} \leftarrow x_{\text{mixed}}[:, : L, :]$             *Truncate to original sequence length $L$*
15: $y \leftarrow \text{Linear}(x_{\text{causal}})$
16: **return** $y$

---

In Algorithm 1, the notation follows standard deep learning conventions. We denote the batch size as $B$, the input sequence length as $L$, and the model's hidden dimension as $D$. The number of parallel heads within the MHF module is denoted by $h$, where the dimension of each head $D_{\text{head}} = D/h$.

A key implementation detail involves the `Conv1d` layers used in lines 2 and 6. Standard deep learning libraries typically expect 1D convolutional inputs in the format $(N, C, L)$, where $N$ is the batch size, $C$ is the number of channels, and $L$ is the sequence length. Therefore, to ensure the convolution is applied along the sequence length dimension, we first permute the input tensor from its shape $\mathbb{R}^{B \times L \times D}$ to $\mathbb{R}^{B \times D \times L}$. Following the convolution, the output is permuted back to $\mathbb{R}^{B \times L \times D}$ to ensure dimensional consistency for subsequent operations.

## B  APPENDIX - STATEMENT ON THE USE OF LARGE LANGUAGE MODELS

Throughout the preparation of this manuscript, we utilized a large language model (LLM) as a general-purpose assistant. The LLM's role was primarily that of a collaborative tool for language refinement, structural organization, and formatting, under the direct guidance and critical supervision of the human authors.

The specific uses of the LLM in the writing process include:

- **Language Refinement:** The LLM assisted in iteratively refining sentence structure, word choice, and overall tone to align with standard academic English. This process involved

numerous cycles of author-led prompting, critical review, and editing to ensure the final text accurately reflected our intended meaning and technical nuances.

- **Figure, Table, Pseudocode and Citations Formatting:** It was used to generate LaTex code to insert figures and tables uploaded by authors. It was also used to convert a PyTorch code implementation of our MHF module into a LaTeX pseudocode algorithm for clarity. Besides, it helped with formatting citations in BibTeX and resolving LaTeX typesetting queries.

It is important to clarify that all core research ideas, the TransFourier architecture design, the conceptual analyses, and the experimental framework are the original intellectual contributions of the human authors. The LLM did not contribute to the research ideation. The authors retained full intellectual control throughout the process, directed all revisions, and assume complete responsibility for the scientific validity and final content of this manuscript.

