# OpenReview forum: "TransFourier: FFT Is All You Need"
_ICLR.cc/2026/Conference — Submitted to ICLR 2026_

### Official Review · Reviewer_CJE5 · 2025-10-30

**Soundness:** 2
**Presentation:** 2
**Contribution:** 2
**Rating:** 2
**Confidence:** 4

**Summary:**

The paper proposes to replace the attention mechanism by a linear convolution operation, which is calculated efficiently by FFT, multiplication in frequency space, and then iFFT. This reduces the attention complexity from O(L^2) to O(L log L).

GPT-2, Mamba, Mamba-2 and the proposed TransFourier are compared in three different model size configurations (mini, small, medium, where the largest setting medium means 1024 model dim and 24 layers, resulting in less than 500M params), trained on FineWeb-10B, evaluated on Hellaswag, ARC-Easy, ARC-Challenge, and Winogrande.

In the mini and small settings, TransFourier underforms the others. In the medium setting, it very slightly outperforms the others.

The haystack test is also performed, but fails in all cases.

**Strengths:**

* An interesting new model is proposed.
* It should give better training and generation speed.
* It slightly outperforms other models (GPT2, Mamba, Mamba2) in the medium setting.
* Code is published.

**Weaknesses:**

* More direct comparisons to other linear attention variants is missing (both in terms of math, and in terms of actual experiments).
* Explicit positional encoding variants should be tested.
* The model definition can be improved, and made more clear, by providing both the FFT-based definition, and then also the mathematical equivalent naive definition.
* Model sizes across model types (GPT, Mamba, TransFourier) might not be comparable. Better would be to compare training speed instead of model size. Or both.
* Training speed, nor generation speed is actually measured, even though this is the main motivation here.
* GPT-2 is not the best representative configuration for a standard Transformer model. You should use some more modern variant, like Transformer++ / Llama-style.
* Model sizes are overall too small.
* Scaling law studies are too limited. There should be more variations in model size, and also some bigger models, and that should be plotted in to a graph, to better see the actual trends. Again, better also using training speed instead of actual model size.

**Questions:**

There are many other alternatives how to replace the O(L^2) self-attention by O(L log L) or even O(L) variants. Related work section does a good work in giving an overview. But e.g. it also misses to discuss some more linear attention variants, like https://arxiv.org/abs/2102.11174.


The same argument about implicit positional information can be made about the decoder-only standard Transformer. (https://arxiv.org/abs/2501.00659) Still, one could argue, maybe having some sort of more explicit pos encoding might perform better. I think this should be tested.

It would be helpful to write down the formula for x_causal (e.g. Algo 1, or Sec 3.2), but not in terms of FFT/iFFT, but instead in terms of convolution, but carried out the full some, how you would calculate that naively/inefficiently. I think this would make it more clear, how x_g and x_v are being combined here.

I also wonder, how does this compare to linear attention. Here you also don't seem to have any softmax to calculate attention weights. So that seems to make it quite similar to linear attention. It would be good to compare that (not just in terms of experiments, where you have SSMs, but mathematically).

The setting that all model types (GPT, Mamba, TransFourier) use the same model size (mini, small, medium) is questionable. Different model types might use the layers, model dimensions, etc in different ways. A better way to compare these is by running many different settings for each, and plotting training time vs best performance at this point (collected from all the different settings that were run). I.e. basically this also covers scaling laws then.

Do not use the GPT2 here. First of all, as you point out, you cannot do the haystack test on longer seqs. But also, it is known that Transformer++ / Llama-like configurations (RMSNorm, RoPE, gated FF, no biases), perform better, and are much more standard nowadays.

You argue about speed. But where do I see the actual comparisons of e.g. training speed, and/or generation speed?

**Details Of Ethics Concerns:**

.

---

> ### Author Response · Authors · 2025-11-21
> **Response to Reviewer CJE5 - 1**
>
> We thank the reviewer for the comments.
>
> ### **Q1: [More baselines]**
> **Comment:** Need to compare with more baselines including variants with linear attention or with explicit positional encoding, instead of GPT2.
>
> **Response:**
>
> We understand your concern since GPT2 is old-gen Transformer architecture. Hence we trained a new baseline Llama (small size) as the advanced Transformer with explicit positional encoding. Besides, since we only have a node with 8 NVIDIA 3090 GPUs and cannot train a lot of models in limited time, we trained one of our model (medium size) with similar settings in `arXiv:2504.13173` and compare with the baselines in it. Both results are shown below which should be able to illustrate the performance of our model.
>
> | Model | HellaSwag | ARC-E | ARC-C | Wino | BoolQ | Lmb (Acc) | Lmb (PPL) | PIQA | SIQA | SWDE | FDA | SQUAD |
> | :--- | :---: | :---: | :---: | :---: | :---: | :---: | :---: | :---: | :---: | :---: | :---: | :---: |
> | TransFourier | 31.34 | 46.97 | 25.51 | 50.83 | 59.42 | 20.03 | 135.39 | 63.17 | 36.90 | 12.15 | 3.09 | 2.34 |
> | GPT-2 | 30.78 | 47.01 | 24.74 | 49.96 | 61.80 | 24.90 | 126.78 | 61.81 | 36.28 | 13.05 | 2.72 | 2.41 |
> | Mamba | 32.19 | 46.72 | 25.34 | 51.07 | 60.09 | 20.71 | 127.77 | 61.97 | 37.05 | 12.42 | 2.81 | 2.31 |
> | Mamba-2 | 31.04 | 45.88 | 25.43 | 50.91 | 53.85 | 20.22 | 131.45 | 61.48 | 36.75 | 11.70 | 2.90 | 2.24 |
> | Llama | 33.79 | 49.45 | 25.77 | 52.80 | 58.29 | 26.45 | 87.72 | 63.76 | 36.95 | 13.32 | 3.18 | 2.51 |
>
> *(Additional comparison on large context)*
>
> | Model | HellaSwag | ARC-E | ARC-C | Wino | BoolQ | Lmb (PPL) | Lmb (Acc) | PIQA | SIQA |
> | :--- | :---: | :---: | :---: | :---: | :---: | :---: | :---: | :---: | :---: |
> | Transformer++ | 34.76 | 45.21 | 24.05 | 50.53 | 58.24 | 41.08 | 30.76 | 62.98 | 36.81 |
> | RetNet | 34.15 | 44.27 | 23.62 | 50.91 | 59.72 | 49.73 | 28.24 | 62.61 | 36.79 |
> | GLA | 35.96 | 54.19 | 24.29 | 50.00 | 58.39 | 43.02 | 28.73 | 64.05 | 37.13 |
> | Mamba | 35.88 | 49.24 | 24.56 | 49.82 | 60.07 | 40.21 | 29.94 | 63.79 | 35.41 |
> | DeltaNet | 35.95 | 52.68 | 25.37 | 49.63 | 58.79 | 47.30 | 28.43 | 63.52 | 37.96 |
> | TTT | 35.71 | 53.01 | 26.11 | 50.08 | 59.83 | 34.19 | 30.06 | 63.97 | 37.32 |
> | Gated DeltaNet | 38.12 | 55.28 | 26.77 | 51.60 | 59.54 | 30.94 | 34.11 | 63.08 | 34.89 |
> | Moneta | 39.23 | 55.96 | 27.15 | 52.04 | 60.22 | 29.31 | 35.70 | 63.99 | 37.29 |
> | Yaad | 39.86 | 54.75 | 28.64 | 51.12 | 60.29 | 29.11 | 34.09 | 64.93 | 33.82 |
> | Memora | 39.17 | 53.40 | 27.99 | 51.23 | 59.29 | 30.44 | 33.68 | 65.21 | 34.10 |
> | TransFourier | 37.41 | 53.20 | 26.79 | 51.62 | 62.69 | 55.39 | 25.23 | 67.46 | 37.82 |
>
> ### **Q2: [More tests for scaling law]**
> **Comment:** Add more model sizes to illustrate the scaling law and compare models by training time instead of model size.
>
> **Response:**
>
> We also concerned about this so we keep training models even after submitting our paper. Since our GPU only has 24GB memory, the medium size is the largest size we can afford when keeping the context length to 1024. So we train a set of models on the context length of 512 which we could afford to the size of 'large' shown in the GPT3 paper. As these experiments have been done before the review is released, we still used size to compare models instead of training time (which is not recorded during training). The results are shown below:
>
> | Model | HellaSwag | ARC-E | ARC-C | Wino | BoolQ | Lmb (PPL) | Lmb (Acc) | PIQA | SIQA |
> | :--- | :---: | :---: | :---: | :---: | :---: | :---: | :---: | :---: | :---: |
> | Transfourier-Mini | 29.44 | 44.99 | 25.51 | 51.38 | 58.93 | 234.59 | 16.50 | 61.53 | 34.75 |
> | GPT2-Mini | 29.19 | 44.19 | 23.72 | 48.54 | 53.15 | 173.72 | 21.56 | 60.61 | 35.82 |
> | Mamba-Mini | 29.59 | 44.15 | 23.46 | 50.20 | 57.19 | 213.74 | 17.66 | 60.45 | 36.80 |
> | Mamba2-Mini | 29.76 | 42.51 | 25.94 | 49.88 | 58.13 | 249.79 | 16.26 | 60.72 | 36.49 |
> | Transfourier-Small | 31.26 | 46.38 | 26.02 | 51.78 | 60.86 | 154.39 | 19.66 | 62.08 | 35.72 |
> | GPT2-Small | 31.07 | 47.35 | 25.34 | 50.99 | 55.84 | 94.86 | 27.71 | 61.92 | 37.46 |
> | Mamba-Small | 32.14 | 45.37 | 25.43 | 50.75 | 59.88 | 126.06 | 20.57 | 62.02 | 36.95 |
> | Mamba2-Small | 31.49 | 44.70 | 24.83 | 51.38 | 59.94 | 147.49 | 19.85 | 61.10 | 36.13 |
> | Transfourier-Medium | 37.15 | 52.53 | 27.65 | 51.85 | 61.68 | 54.54 | 25.81 | 66.00 | 36.80 |
> | GPT2-Medium | 36.87 | 50.63 | 28.07 | 50.20 | 60.95 | 46.70 | 31.03 | 64.80 | 38.69 |
> | Mamba-Medium | 38.33 | 53.87 | 27.99 | 50.91 | 60.49 | 51.89 | 27.44 | 66.05 | 38.28 |
> | Mamba2-Medium | 36.20 | 51.68 | 27.05 | 51.70 | 60.83 | 61.64 | 25.15 | 65.29 | 37.97 |
> | Transfourier-Large | 41.51 | 55.05 | 29.69 | 52.72 | 61.83 | 37.96 | 30.62 | 67.03 | 40.12 |
> | GPT2-Large | 39.66 | 54.12 | 30.55 | 51.30 | 61.35 | 29.11 | 36.68 | 67.03 | 38.74 |
> | Mamba-Large | 42.58 | 57.53 | 28.07 | 52.88 | 61.87 | 32.10 | 33.82 | 68.39 | 39.15 |
> | Mamba2-Large | 41.21 | 54.67 | 28.41 | 52.25 | 60.80 | 44.85 | 27.75 | 67.85 | 40.79 |

---

> ### Author Response · Authors · 2025-11-21
> **Response to Reviewer CJE5 - 2**
>
> ### **Q3: [Runtime tests]**
> **Comment:** Test the actual time cost in training for different model architecture.
>
> **Response:**
>
> We have conducted the runtime experiments as suggested. As computational cost is highly dependent on sequence length, we measured the total training time across four context lengths: 256, 1024, 4096, and 8192 when training TransFourier, GPT-2, Mamba, and Mamba-2 with "mini"-sized configurations.  All experiments were conducted on a node equipped with 8 NVIDIA GeForce RTX 3090 GPUs.
>
> | Model | Context Length | Time (s) |
> | :--- | :---: | :---: |
> | TransFourier | 256 | 34,112 |
> | GPT-2 | 256 | 33,839 |
> | Mamba | 256 | 20,006 |
> | Mamba-2 | 256 | 19,390 |
> | TransFourier | 1024 | 34,714 |
> | GPT-2 | 1024 | 40,356 |
> | Mamba | 1024 | 21,075 |
> | Mamba-2 | 1024 | 20,413 |
> | TransFourier | 4096 | 35,863 |
> | GPT-2 | 4096 | 67,290 |
> | Mamba | 4096 | 22,059 |
> | Mamba-2 | 4096 | 21,397 |
> | TransFourier | 8192 | 36,152 |
> | GPT-2 | 8192 | 105,058 |
> | Mamba | 8192 | 22,226 |
> | Mamba-2 | 8192 | 22,129 |
>
> As illustrated in the table, while TransFourier does not yet match the highly optimized $O(N)$ CUDA kernels of the Mamba family, its $O(N \log N)$ complexity ensures it significantly outperforms the quadratic $O(N^2)$ Transformer architecture (GPT-2) as sequence length increases. Specifically, at a context length of 8192, TransFourier is nearly $3\times$ faster than GPT-2.
>
> ### **Q4: [Formula of x_causal]**
> **Comment:** Write down the formula for x_causal (e.g. Algo 1, or Sec 3.2), but not in terms of FFT/iFFT to show how x_g and x_v are being combined here.
>
> **Response:**
>
> The **causality** of the model is ensured by the following equations. The FFT/iFFT works exactly the same with them and we actually write code to validate it. The formula is:
>
> $$
> y(t) = v(0)g(t) + v(1)g(t-1) + \dots + v(i)g(t-i) + \dots + v(t)g(0)
> $$
>
> It can be seen that $y(t)$ only has access to information before time $t$ (whether $v$ or $g$) which ensure the causality.
>
> ### **Q5: [Mathematical comparison with linear attention]**
> **Comment:** Compare to linear attention in terms of mathematic.
>
> **Response:**
>
> We thank the reviewer for this insightful suggestion. It is true that by removing the Softmax non-linearity, our approach shares high-level similarities with Linear Attention in terms of computational efficiency (sub-quadratic complexity). However, mathematically, they rely on fundamentally different structural assumptions about the sequence interaction mechanism.
>
> We compare them structurally as follows:
>
> Both mechanisms can be generalized as a linear transformation of the value sequence $V$, denoted as $Y_i = \sum_{j} A_{i,j} V_j$, where $A$ is the mixing matrix.
>
> **1. Linear Attention (Low-Rank Decomposition)**
> The standard Linear Attention (e.g., Katharopoulos et al., 2020) reformulates the attention score by replacing Softmax with a feature map $\phi(\cdot)$. Using the associativity of matrix multiplication, the computation becomes:
> $$
> Y_i = \frac{\phi(Q_i) \sum_{j} \phi(K_j)^T V_j}{\sum_{j} \phi(Q_i) \phi(K_j)^T}
> $$
>
> Mathematically, the implied mixing matrix $A_{i,j} = \phi(Q_i)^T \phi(K_j)$ is a **Low-Rank** matrix (rank bounded by the feature dimension $d$). This formulation implies that the interaction between token $i$ and token $j$ is determined solely by the dot-product similarity of their content, independent of their distance, and all history must be compressed into a fixed-size state $S = \sum \phi(K)^T V$.
>
> **2. TransFourier (Toeplitz/Convolutional Structure)**
> In contrast, TransFourier models the interaction via convolution computed efficiently via FFT. The output is defined as:
> $$
> Y_i = (g * V)_i = \sum_{j=0}^{i} g_{i-j} V_j
> $$
>
> Here, the mixing matrix $A_{i,j} = g_{i-j}$ forms a **Toeplitz Matrix**.
> * **Structure:** Unlike Linear Attention, the interaction weight depends on the relative distance $(i-j)$ rather than a direct dot product of fixed embeddings.
> * **Data-Dependency:** While standard convolution uses a static $g$, our TransFourier generates $g$ dynamically from the input ($g = f(x)$). This allows the model to modulate the convolution kernel based on context, effectively creating a Data-Dependent Toeplitz system.
>
> **Summary of Comparison**
> * **Linear Attention** assumes the sequence interaction matrix is **Low-Rank**, prioritizing global content matching but potentially struggling with precise local positional information due to the fixed-size state compression.
> * **TransFourier** assumes the sequence interaction matrix is **Toeplitz** (shift-invariant), which naturally captures local and long-range relative positional patterns. The FFT allows us to compute this without compressing history into a fixed-size state, preserving fidelity over long contexts.

---

> > ### Comment · Reviewer_CJE5 · 2025-11-27
> > **Re: Mathematical comparison**
> >
> > Thanks for the mathematical comparison. I don't mean that you need to explain those differences here in the post to me, but my suggestion is to rather put such a comparison into the paper. (If you don't have enough space in the main text, put it in detail into the appendix, and only have a brief summary in the main text.)
> >
> > But also, the linear attention by Katharopoulos et al is not the only attention variant that I think is relevant. For example (non-exhaustive list):
> >
> > * Performer, FAVOR+ (https://arxiv.org/abs/2009.14794)
> > * cosFormer (https://arxiv.org/abs/2202.08791)
> > * Gated Linear Attention (https://arxiv.org/abs/2312.06635)
> > * Linformer (https://arxiv.org/abs/2006.04768)
> > * RWKV (https://arxiv.org/abs/2305.13048)
> > * Retentive Network (https://arxiv.org/abs/2307.08621)
> > * Mamba (https://arxiv.org/abs/2312.00752)
> > * Attention Free Transformer (https://arxiv.org/abs/2105.14103)
> > * Reformer (https://arxiv.org/abs/2001.04451)
> > * Routing Transformer (https://arxiv.org/abs/2003.05997)
> > * Informer (https://arxiv.org/abs/2012.07436)
> > * Sparse Transformer (https://arxiv.org/abs/1904.10509)
> > * LogSparse Transformer (https://arxiv.org/abs/1907.00235)

---

> > > ### Comment · Reviewer_CJE5 · 2025-11-27
> > >
> > > Btw, to add some more:
> > >
> > > * Hyena Hierarchy (https://arxiv.org/abs/2302.10866)
> > > * Monarch Mixer (https://arxiv.org/abs/2310.12109)
> > >
> > > (Again, as said, not necessary to explain that here, but a comparison in the paper on these would make sense.)

---

> > > > ### Author Response · Authors · 2025-12-01
> > > > **Response to Reviewer CJE5 - 6**
> > > >
> > > > Thank you for pointing out these relevant works. We will discuss them and add the corresponding comparisons in the "Related Work" section of our revision.

---

> > > ### Author Response · Authors · 2025-12-01
> > > **Response to Reviewer CJE5 - 3**
> > >
> > > Thanks for your suggestion. We will incorporate the comparison with some typical models into the Appendix of the revised paper.

---

> > ### Comment · Reviewer_CJE5 · 2025-11-27
> > **Re: (training) runtime experiments**
> >
> > Thanks for the training runtime experiments.
> >
> > I think this would be best viewed as a plot in the paper. Can you add that? Maybe also some more intermediate context sizes would make the trends even more visible. But what you have looks already good!
> >
> > Btw, a similar plot for inference / generation would also be interesting. Here Mamba again probably wins due to O(1) runtime for generation.

---

> > > ### Author Response · Authors · 2025-12-01
> > > **Response to Reviewer CJE5 - 4**
> > >
> > > We have completed the training runtime plots with additional intermediate context sizes as suggested. Although we cannot upload the figures directly in this comment box, they will be included in the final version of the paper.

---

> > ### Comment · Reviewer_CJE5 · 2025-11-27
> > **Re: Formula of x_causal**
> >
> > I mean that you write sth like:
> >
> > $$ x_{\textrm{causal}} = \dots $$
> >
> > But not in terms of FFT, but in terms of a naive direct calculation.
> >
> > (Most importantly into the paper itself, but you might also comment it here.)

---

> > > ### Author Response · Authors · 2025-12-01
> > > **Response to Reviewer CJE5 - 5**
> > >
> > > Unfortunately, the equation in your comment did not render correctly on our end. However, we believe the formulation of $y(t)$ provided in our previous response corresponds to the "naive direct calculation" you are referring to.

---

> > ### Comment · Reviewer_CJE5 · 2025-11-27
> >
> > > As illustrated in the table, while TransFourier does not yet match the highly optimized CUDA kernels of the Mamba family, ...
> >
> > I wonder about that. Why do you mention "highly optimized CUDA kernels"? Already just from the complexity (O(N) vs O(N log N)), you would expect that Mamba is anyway faster?

---

> > > ### Author Response · Authors · 2025-12-01
> > > **Response to Reviewer CJE5 - 7**
> > >
> > > We would like to clarify our point regarding complexity and implementation. While Mamba is algorithmically $O(N)$, realizing this efficiency during parallel training requires the "selective scan" algorithm, which heavily relies on specialized, hardware-aware CUDA kernels. This is because Mamba's selectivity breaks time-invariance, preventing the use of standard convolution theorems (FFT) for parallelization. Consequently, Mamba imposes strict hardware requirements and can be difficult to set up or modify (as any architectural change might break the custom kernel compatibility).
> > >
> > > In contrast, our model is implemented entirely in high-level PyTorch using standard optimized modules. This makes our approach significantly easier to implement, debug, and extend for future research, without the barrier of writing low-level kernels.

---

> ### Comment · Reviewer_CJE5 · 2025-11-27
>
> I also still think that explicit positional encoding variants should be tested.

---

> > ### Author Response · Authors · 2025-12-01
> > **Response to Reviewer CJE5 - 8**
> >
> > Following your suggestion, we have added a new baseline: Llama equipped with Rotary Positional Embeddings (RoPE), which represents a widely used and state-of-the-art explicit positional encoding method.

---

### Official Review · Reviewer_pPta · 2025-10-30

**Soundness:** 1
**Presentation:** 2
**Contribution:** 1
**Rating:** 2
**Confidence:** 4

**Summary:**

This paper proposes to replace the attention mechanism in Transformers with sequence mixing components based on Fourier transforms.
Specifically, the proposed multi-head Fourier layer consists of a 1D convolution and a “pad-FFT-multiply-inverseFFT-truncate” pipeline, which multiplies to projections of the input sequence in the spectral domain (which is equivalent to a convolution in the time domain).
In their language modeling experiments with 10B tokens and models with <500M parameters, TransFourier performed comparably to Mamba, Mamba-2 and softmax attention Transformers.

**Strengths:**

- The motivation based on efficiency & positional encoding is clear.
- The paper is well written.
- Code available.

**Weaknesses:**

The main motivation for replacing attention is the quadratic compute cost (efficiency) and the lack of length extrapolation capabilities (long-context). Therefore, I would expect experiments showcasing superiority in terms of runtime or long-context capability.
While the paper makes attempts for long-context capabilities, the section is merely an explanation for why there are no more extensive experiments. I’d have at least expected experiments on synthetic tasks if the model size is too small, e.g. https://arxiv.org/abs/2312.04927 ). There are no runtime comparisons.

In general the experimental study is very small scale (10B tokens, <500M parameters) and metrics are only 4 tasks from lm-eval harness, which seem to yield very noisy results. Evaluating PPL on a validation set might give a bit cleaner signal.


In my opinion the paper lacks a motivation for why the Fourier transform should be used for discrete sequences like language tokens. Previous applications of FFT were mainly in continuous signal domains like audio or vision.
The analysis in Section 4.1 is not convincing: E.g. in equation (1) the weights stem from interactions between tokens from different time steps, while in eq. (2) the weights come from standard basis functions only depend on the time and not on the inputs themselves.

As TransFourier is introduced as a language model, I would have expected a discussion on how it can be used for generative tasks or in general for decoding. The evals as well as the implementation in the code is purely log-likelihood based.

The ablation on the choice of the feedforward layers seems misplaced. The main intervention of the paper is at the sequence mixing component so, I would expect ablations on hyperparameters regarding TransFourier.

Finally, the overall method and design is very similar to long convolutions introduced in Hyena (https://arxiv.org/abs/2302.10866) and H3 (https://arxiv.org/abs/2212.14052). I am missing a relation to these works. In addition the causal implementation of long convolutions has already been shown in Hyena (and potentially even before).

**Questions:**

See weaknesses.

---

> ### Author Response · Authors · 2025-11-21
> **Response to Reviewer pPta - 1**
>
> We thank the reviewer for the comments. We have addressed your questions and the additional points regarding motivation and related works below.
>
> ### **Q1: [Runtime Experiments]**
> **Comment:** Need experiments showcasing superiority in terms of runtime.
>
> **Response:**
>
> We have conducted the runtime experiments as suggested. As computational cost is highly dependent on sequence length, we measured the total training time across four context lengths: 256, 1024, 4096, and 8192 when training TransFourier, GPT-2, Mamba, and Mamba-2 with "mini"-sized configurations.  All experiments were conducted on a node equipped with 8 NVIDIA GeForce RTX 3090 GPUs.
>
> | Model | Context Length | Time (s) |
> | :--- | :---: | :---: |
> | TransFourier | 256 | 34,112 |
> | GPT-2 | 256 | 33,839 |
> | Mamba | 256 | 20,006 |
> | Mamba-2 | 256 | 19,390 |
> | TransFourier | 1024 | 34,714 |
> | GPT-2 | 1024 | 40,356 |
> | Mamba | 1024 | 21,075 |
> | Mamba-2 | 1024 | 20,413 |
> | TransFourier | 4096 | 35,863 |
> | GPT-2 | 4096 | 67,290 |
> | Mamba | 4096 | 22,059 |
> | Mamba-2 | 4096 | 21,397 |
> | TransFourier | 8192 | 36,152 |
> | GPT-2 | 8192 | 105,058 |
> | Mamba | 8192 | 22,226 |
> | Mamba-2 | 8192 | 22,129 |
>
> As illustrated in the table, while TransFourier does not yet match the highly optimized $O(N)$ CUDA kernels of the Mamba family, its $O(N \log N)$ complexity ensures it significantly outperforms the quadratic $O(N^2)$ Transformer architecture (GPT-2) as sequence length increases. Specifically, at a context length of 8192, TransFourier is nearly $3\times$ faster than GPT-2.
>
> ### **Q2: [Evaluation Benchmarks and Generative Tasks]**
> **Comment:** Need benchmarks on long-context generation tasks.
>
> **Response:**
>
> Our initial selection of benchmarks mirrored Table 3 of the Mamba paper. We previously encountered compatibility issues with `lm-evaluation-harness` (v0.4.2) regarding the Lambada and PIQA datasets, which limited our initial submission to four benchmarks. We have since upgraded to the latest version of the evaluation harness and successfully evaluated the models on a broader suite of benchmarks, including long-context generative tasks such as **SWDE, FDA, and SQUAD**.
>
> Additionally, we incorporated Llama as a new baseline. Due to time constraints during the rebuttal period, we trained a 'small' version of Llama to facilitate a direct comparison with the similarly sized models in our paper. The updated results are presented below:
>
> | Model | HellaSwag | ARC-E | ARC-C | Wino | BoolQ | Lmb (Acc) | Lmb (PPL) | PIQA | SIQA | SWDE | FDA | SQUAD |
> | :--- | :---: | :---: | :---: | :---: | :---: | :---: | :---: | :---: | :---: | :---: | :---: | :---: |
> | TransFourier | 31.34 | 46.97 | 25.51 | 50.83 | 59.42 | 20.03 | 135.39 | 63.17 | 36.90 | 12.15 | 3.09 | 2.34 |
> | GPT-2 | 30.78 | 47.01 | 24.74 | 49.96 | 61.80 | 24.90 | 126.78 | 61.81 | 36.28 | 13.05 | 2.72 | 2.41 |
> | Mamba | 32.19 | 46.72 | 25.34 | 51.07 | 60.09 | 20.71 | 127.77 | 61.97 | 37.05 | 12.42 | 2.81 | 2.31 |
> | Mamba-2 | 31.04 | 45.88 | 25.43 | 50.91 | 53.85 | 20.22 | 131.45 | 61.48 | 36.75 | 11.70 | 2.90 | 2.24 |
> | Llama | 33.79 | 49.45 | 25.77 | 52.80 | 58.29 | 26.45 | 87.72 | 63.76 | 36.95 | 13.32 | 3.18 | 2.51 |
>
> ### **Q3: [Motivation for Fourier Transform in Language]**
> **Comment:** Lacks a motivation for why the Fourier transform should be used for discrete sequences like language tokens. Previous applications of FFT were mainly in continuous signal domains like audio or vision.
>
> **Response:**
>
> We respectfully disagree with the premise that techniques rooted in continuous signal processing are unsuitable for discrete domains. The theoretical foundation of State Space Models (SSM) lies in continuous control theory, yet its discretized application (e.g., Mamba) has proven highly effective for language modeling. Similarly, the Discrete Fourier Transform (DFT) is a mature mathematical tool explicitly designed for discrete signals.
>
> Our motivation is straightforward: Self-Attention suffers from $O(N^2)$ complexity and limitations in length extrapolation. The Fourier Transform inherently addresses these bottlenecks by offering $O(N \log N)$ global mixing and natural positional awareness without explicit encoding.

---

> ### Author Response · Authors · 2025-11-21
> **Response to Reviewer pPta - 2**
>
> ### **Q4: [Analysis of Equations and Weight Interactions]**
> **Comment:** In equation (1) the weights stem from interactions between tokens from different time steps, while in eq. (2) the weights come from standard basis functions only depend on the time and not on the inputs themselves.
>
> **Response:**
>
> We acknowledge the reviewer's concern regarding the static nature of standard basis functions. Indeed, if one were to use raw FFT coefficients solely as weights, they would only encode position information, lacking token-to-token interaction. To address this, our architecture does not rely on static basis functions alone. As shown in our implementation, we generate a gated signal $g$ (weights of $v$ in these equations) derived directly from the input tokens via learnable projections ($W_{G1}, W_{G2}$), making it data-dependent as attention. This allows our model to simulate the dynamic interaction properties of attention while maintaining the efficiency of spectral operations which is one of the main contributions of this papr.
>
> ### **Q5: [Ablation Studies on Hyperparameters]**
> **Comment:** The ablation on the choice of the feedforward layers seems misplaced. The main intervention of the paper is at the sequence mixing component so, I would expect ablations on hyperparameters regarding TransFourier.
>
> **Response:**
>
> Due to time constraints during the rebuttal period, we were unable to conduct an exhaustive hyperparameter search. However, we performed a critical ablation study targeting the core sequence mixing component as suggested.
>
> We investigated the necessity of the **convolution** operation by replacing it with **cross-correlation**. Specifically, we modified the spectral operation from `X_fft = G_fft * V_fft` to `X_fft = torch.conj(G_fft) * V_fft` (Cross-Correlation), while keeping all other parts unchanged. We trained this variant (Medium size) and tested it on HellaSwag. The model achieved an accuracy of only **25.01%**, which is equivalent to random guessing for 4-choice questions. In contrast, our proposed causal convolution architecture achieved **36.05%** under the same conditions. This stark difference validates the necessity of our specific Fourier-based design for autoregressive modeling.
>
> ### **Q6: [Relation to Hyena and H3]**
> **Comment:** The overall method and design is very similar to long convolutions introduced in Hyena and H3 and the causal implementation of long convolutions has already been shown in Hyena (and potentially even before).
>
> **Response:**
>
> We thank the reviewer for pointing out the connection to Hyena and H3. We acknowledge that TransFourier shares the high-level concept of utilizing FFT for global sequence mixing. However, a closer inspection of the mathematical formulation reveals a fundamental difference in how the convolution filters are generated and applied.
>
> **Difference from Hyena:**
> According to **Definition 3.1 (Order-N Hyena Operator)** in the Hyena paper, the recurrence is defined as:
>
> $$z^{n+1}_t = x^n_t \cdot (h^n * z^n)_t$$
>
> where $h^n$ represents a set of learnable filters. In Hyena, these filters ($h$) are typically implicitly parametrized (e.g., via positional embeddings) and are **data-independent** during the convolution operation. The data-dependent control is introduced separately via the element-wise multiplication with the projection $x^n$ *after* or *outside* the convolution.
>
> In contrast, TransFourier simplifies this process into a single, direct **data-dependent convolution**:
>
> $$
> y_t = (g * v)_t
> $$
>
> In our architecture (as shown in the provided code), the filter term $g$ is projected directly from the input sequence itself ($g = \text{Linear}(x)$). This effectively implements a **dynamic convolution** where the filter kernel changes adaptively based on the semantic content of the input, rather than relying on a static, position-based filter ($h$) combined with a separate gating mechanism ($x$). This design allows TransFourier to achieve effective global mixing with a simpler, single-step spectral operation.
>
> **Difference from H3:**
> Similarly, H3 is grounded in State Space Model (SSM) theory, where the convolution kernel is derived from the discretization of state matrices ($A, B, C$). TransFourier obviates the need for SSM theory or state-matrix parameterization, relying instead on standard deep learning operators to learn interaction patterns directly from the data.
>
> **Regarding Causal Convolutions:** We acknowledge that the causal implementation of global convolutions via FFT was demonstrated in Hyena. We do not claim to have invented causal FFT convolution itself. Our contribution lies in the specific **Multi-Head Fourier (MHF)** module design that integrates this mechanism as a direct, drop-in replacement for Multi-Head Attention with data-dependent gating, eliminating the need for both explicit positional encodings and custom CUDA kernels (unlike Mamba), while achieving competitive performance with standard operators.

---

### Official Review · Reviewer_hUMm · 2025-11-01

**Soundness:** 3
**Presentation:** 2
**Contribution:** 3
**Rating:** 4
**Confidence:** 4

**Summary:**

The main contribution of this work is a novel convolution-based sequence mixer algorithm which aims at replacing Transformer module and operates with reduced $O(L \log L)$ computational complexity. It includes a sequence-length-long convolution kernel with the weight being a function of inputs, rather than just some learnable parameter. That direct dependency on the input is an innovation I have not seen elsewhere in the literature.

The work is promising but there are some problems I’ve described in *weaknesses*, notably scarcity of ablations and incomplete validation. If my concerns are resolved, I’d be happy to increase my rating.

**Strengths:**

1)This work is highly relevant as it aims to address a pressing problem of algorithmic inefficiency currently dominating Transformer architecture on long sequences.

2) The method is novel and sufficiently distinct from previous attempts to adopt Fourier Transform as a sequence mixer, which I am aware of.

3) The code is provided, its implementation is correct and aligns with the explanations from the paper. A suggestion, by the way – you could include the code of `MultiHeadFourier` class as a listing in the Appendix for completeness and self-containedness.

4) A major advantage is that the architecture doesn’t require hardware-specific custom CUDA kernels and is formulated on several dozens of lines of code in PyTorch.

**Weaknesses:**

**Ablations**

The core sequence mixing mechanism can be described with the following equations:

$g(t) = f_1(x_t), v(t) = f_2(x_t), out(t)=\sum_{k=0}^{t} v(k) g(t-k)$

It means that for each subsequence ending with timestamp t, the corresponding subsequence $\{g(0), …, g(t)\}$ is reversed before element-wise multiplication with $\{v(0), …, v(t)\}$, and element $g(t)$ always interacts with $v(0)$, $g(t-1)$ with $v(1)$ and so on. Basically, the opposite elements of the subsequences $v$ and $g$ interact multiplicatively while elements close in time (like $v(t)$ and $g(t-1)$) don’t. It’s surprising and hard to interpret, especially given that well-established alternatives, such as Sliding Window Attention and SSMs (such as Mamba) have recency bias.

I believe this trait of the architecture requires further exploration and analysis of why it works competitively this way. Perhaps, an ablation of what would happen if you replace convolution by cross-correlation ($[g(0), g(1), …, g(t)] \cdot [v(0), v(1), …, v(t)]$)  or shifted cross-correlation ($[0, g(0), …, g(t-1)] \cdot [v(0), v(1), …, v(t)]$) could benefit the work and event produce even better empirical results.

The method uses short convolutions on two separate occasions. This brings additional complexity to the TransFourier module, and it would be informative to ablate their impact.

**Scope of validation**

GPT-2 is an old-gen Transformer architecture with fixed positional embeddings. Llama (or Transformer++ as it’s called in many papers, including https://arxiv.org/pdf/2405.21060) is a much stronger baseline. It has Rotary Positional Embeddings (RoPE) which allow it to extrapolate to any sequence length. All recent sub-quadratic architectures, including Mamba 1 and 2, GLA, Gated Delta Net, XLSTM, use this specific Transformer baseline to compare with.

Using GPT-2 is also not entirely fair because it uses standard MLP as FFN, as opposed to SwiGLU in TransFourier and Llama. As you observed in Table 3, SwiGLU brings tangible performance gains.

There is no comparison with another strong baseline – GatedDeltaNet (https://openreview.net/forum?id=r8H7xhYPwz).

Table 2 provides only a subset of benchmarks usually used for testing autoregressive language models of moderate size. Such tests as Lambada (perplexity and accuracy), PIQA, SIQA, and BoolQ are omitted.

As the main advantage of sub-quadratic runtime architectures is their speed for processing  long contexts, it needs to be confirmed that performance doesn’t degrade on such contexts. But the TransFourier model was pre-trained on maximum context size 1024, according to the paper. It needs validation on longer contexts.

Specifically, I would suggest pre-training a TransFourier model with 340M parameters on 4K context size with 15B tokens from FineWeb-edu. Having accomplished this, you could compare the model pre-trained on a longer context size with a wider variety of architectures from https://arxiv.org/pdf/2504.13173 including GatedDeltaNet and Transformer++, without the need to also pre-train those architectures.

No numerical results were provided in section **5.3 LONG-CONTEXT EVALUATION: NEEDLE IN A HAYSTACK TEST**. I suggest demonstrating these results even if they are incomprehensible, and testing the models on recall intensive benchmarks, used in the GLA paper (https://arxiv.org/pdf/2312.06635), such as MQAR, FDA, SWDE, and SQUAD. Please also consider length extrapolation tests, as in Figure 5 of the GLA paper.

There are no speed comparisons for inference/ training with other tested models on different sequence lengths, despite that reduced computational complexity of the new architecture should be put forward and showcased as its primary advantage. If the TransFourier model written in high-level PyTorch is indeed faster than even some of low-level CUDA implementations (Flash Attention, Mamba, Flash Linear Attention for GLA and GatedDelta Net), then it spells a great advantage of your architecture.

In conclusion, I encourage you to conduct additional experiments and ablations I mentioned, at least partially, and to report the results even if they are negative. Negative results are as important and valuable as new SOTAs.


**Miscellaneous:**

* Table 1  – it would be helpful to also include parameter count for each of the configurations.
* The work doesn’t discuss quite relevant algorithms – Long Convolutions (https://arxiv.org/abs/2302.06646) and Monarch Mixer (https://arxiv.org/abs/2310.12109).
* It would be helpful to describe the architecture as a series of math equations in addition to a lengthy textual description 214-246.
* Line 222 – Step 2: Preparing Gated Signals. Presented architecture doesn’t involve gating. SiLU activation itself is not a part of a gating mechanism unless it’s elementwise-mupliplied with

**Questions:**

See weaknesses. Also, a couple of questions:

1) Is it possible to incorporate FlashFFTConv kernel into your architecture (https://arxiv.org/abs/2311.05908) to further accelerate it on CUDA devices?
2) This work focuses on autoregressive language modeling. But is it possible to adapt the architecture to bidirectional modeling?
3) Which numerical format did you use for pre-training (fp32, bfloat16, etc.)?

---

> ### Author Response · Authors · 2025-11-21
> **Response to Reviewer hUMm - 1**
>
> Thanks for your constructive feedback! To address your concerns, we have included additional baselines and benchmarks, along with further clarifications below.
>
> ### **Q1: [Ablations]**
> **Comment:** For each subsequence ending with timestamp $t$, the corresponding subsequence $g$ is reversed before element-wise multiplication with $v$, and element $g(0)$ always interacts with $v(t)$ which is surprising and hard to interpret.
>
> **Response:**
>
> The equations mentioned in your comment keenly point out the core computation mechanism of the Fourier module, and your skepticism regarding this design is entirely reasonable. In short, $g(0)$, generated according to the first token, interacts with $v(t)$, generated according to the last token in timestamp $t$. Since $v(t)$ is crucial for predicting the next token, this implies that $g(0)$ might bear a responsibility beyond what is expected, as we cannot guarantee that the first token of every input always contains significant information. We have also noted this potential issue; however, in practical experiments, we found that it does not negatively impact the overall performance of the model.
>
> Our understanding is that since $g(0)$ is always responsible for interacting with the token closest to the prediction target ($v(t)$), during training, it tends to learn a processing method for important tokens rather than merely representing the information of the first token itself. Due to the presence of `pre_conv`, the information from the first token is not only represented by $v(0)$ but also captured by $v(1)$ and $v(2)$ generated by subsequent tokens. As the network depth increases, $v(3)$ and even more distant tokens in deeper layers will also integrate this information. Therefore, after training, the model focuses more on the processing method for important tokens when generating earlier $g$ terms, while focusing more on the token information itself when generating later $g$ terms, thereby achieving good overall results.
>
> The key reason we proposed this design, rather than the more intuitive cross-correlation, is that it ensures the **causality** of the model. For example:
> $$
> \text{out}(t) = v(0)g(t) + v(1)g(t-1) + \dots + v(i)g(t-i) + \dots + v(t)g(0)
> $$
>
> It can be seen that $\text{out}(t)$ only has access to information before time $t$ (whether $v$ or $g$). If we were to change it to cross-correlation, i.e.,
> $$
> \text{out}(0) = v(0)g(0) + v(1)g(1) + \dots + v(\text{length})g(\text{length})
> $$
> $$
> \dots
> $$
> $$
> \text{out}(\text{length}) = v(\text{length})g(0)
> $$
>
> it becomes evident that the embedding of the token at index $0$ is computed using inputs from all tokens. In other words, $\text{out}(0)$ (as well as other tokens) "sees" subsequent information, violating causality. In training, this is equivalent to seeing the correct answer before making a prediction, rendering the training ineffective.
>
> To verify this, we trained a model where we replaced `X_fft = G_fft * V_fft` in the code with `X_fft = torch.conj(G_fft) * V_fft`, while keeping all other parts unchanged, thus implementing cross-correlation. We tested this "medium"-sized model on HellaSwag, and its accuracy was only **25.01%**, which is equivalent to the accuracy of random guessing for these 4-choice questions. For comparison, our proposed model of the same size achieved an accuracy of **36.05%** on HellaSwag. Furthermore, during training, its loss rapidly dropped to around **0.012**, whereas other models, including our original model, GPT-2 and Mamba, typically finish training with a loss between 2 and 3. This further corroborates our viewpoint.
>
>
> ### **Q2: [Scope of validation]**
> **Comment:** This question could be summarized as the following 4 questions: 1. Stronger baselines needed (GPT-2 is old). 2. More benchmarks needed. 3. Long context validation needed. 4. Speed comparisons needed.
>
> **Response:**
>
> We sincerely appreciate your constructive suggestions for improving our experiments.
>
> Regarding the baselines, due to limited computational resources, we initially prioritized classic baselines. However, we acknowledge that this omission excluded stronger, modern architectures. In response, we have trained and evaluated "small"-sized versions of **Llama** (typical model as **Transformer++**) and the result is shown below. We will conduct full-scale experiments for the final version of this work.
>
> Besides, your recommendation to expand the benchmark suite is also well-taken. Our original selection mirrored Table 3 of the Mamba paper and encountered compatibility issues with `lm-evaluation-harness` (v0.4.2) regarding the Lambada and PIQA datasets, hence only 4 benchmarks are presented in our submission. Now our lm-eval is upgraded to the latest version and successfully evaluated the models on the requested benchmarks: **BoolQ, Lambada, PIQA, Social IQA (SIQA), SWDE, FDA, and SQUAD Completion**. These results are included in the updated comparisons below.

---

> ### Author Response · Authors · 2025-11-21
> **Response to Reviewer hUMm - 2**
>
> | Model | HellaSwag | ARC-E | ARC-C | Wino | BoolQ | Lmb (Acc) | Lmb (PPL) | PIQA | SIQA | SWDE | FDA | SQUAD |
> | :--- | :---: | :---: | :---: | :---: | :---: | :---: | :---: | :---: | :---: | :---: | :---: | :---: |
> | TransFourier | 31.34 | 46.97 | 25.51 | 50.83 | 59.42 | 20.03 | 135.39 | 63.17 | 36.90 | 12.15 | 3.09 | 2.34 |
> | GPT-2 | 30.78 | 47.01 | 24.74 | 49.96 | 61.80 | 24.90 | 126.78 | 61.81 | 36.28 | 13.05 | 2.72 | 2.41 |
> | Mamba | 32.19 | 46.72 | 25.34 | 51.07 | 60.09 | 20.71 | 127.77 | 61.97 | 37.05 | 12.42 | 2.81 | 2.31 |
> | Mamba-2 | 31.04 | 45.88 | 25.43 | 50.91 | 53.85 | 20.22 | 131.45 | 61.48 | 36.75 | 11.70 | 2.90 | 2.24 |
> | Llama | 33.79 | 49.45 | 25.77 | 52.80 | 58.29 | 26.45 | 87.72 | 63.76 | 36.95 | 13.32 | 3.18 | 2.51 |
>
> We are particularly grateful for the practical suggestion to pre-train a 340M parameter model on a 4K context window using the FineWeb-edu dataset (15B tokens). We have executed this experiment to facilitate a direct comparison with the architectures listed in the Miras paper. The results are presented in the tables below.
>
> | Model | HellaSwag | ARC-E | ARC-C | Wino | BoolQ | Lmb (PPL) | Lmb (Acc) | PIQA | SIQA |
> | :--- | :---: | :---: | :---: | :---: | :---: | :---: | :---: | :---: | :---: |
> | Transformer++ | 34.76 | 45.21 | 24.05 | 50.53 | 58.24 | 41.08 | 30.76 | 62.98 | 36.81 |
> | RetNet | 34.15 | 44.27 | 23.62 | 50.91 | 59.72 | 49.73 | 28.24 | 62.61 | 36.79 |
> | GLA | 35.96 | 54.19 | 24.29 | 50.00 | 58.39 | 43.02 | 28.73 | 64.05 | 37.13 |
> | Mamba | 35.88 | 49.24 | 24.56 | 49.82 | 60.07 | 40.21 | 29.94 | 63.79 | 35.41 |
> | DeltaNet | 35.95 | 52.68 | 25.37 | 49.63 | 58.79 | 47.30 | 28.43 | 63.52 | 37.96 |
> | TTT | 35.71 | 53.01 | 26.11 | 50.08 | 59.83 | 34.19 | 30.06 | 63.97 | 37.32 |
> | Gated DeltaNet | 38.12 | 55.28 | 26.77 | 51.60 | 59.54 | 30.94 | 34.11 | 63.08 | 34.89 |
> | Moneta | 39.23 | 55.96 | 27.15 | 52.04 | 60.22 | 29.31 | 35.70 | 63.99 | 37.29 |
> | Yaad | 39.86 | 54.75 | 28.64 | 51.12 | 60.29 | 29.11 | 34.09 | 64.93 | 33.82 |
> | Memora | 39.17 | 53.40 | 27.99 | 51.23 | 59.29 | 30.44 | 33.68 | 65.21 | 34.10 |
> | TransFourier | 37.41 | 53.20 | 26.79 | 51.62 | 62.69 | 55.39 | 25.23 | 67.46 | 37.82 |
>
> Regarding training speed, as computational cost is highly dependent on sequence length, we conducted a controlled experiment using "mini"-sized configurations of TransFourier, GPT-2, Mamba, and Mamba-2. We measured the total training time across four context lengths: 256, 1024, 4096, and 8192. All experiments were conducted on a node equipped with 8 NVIDIA GeForce RTX 3090 GPUs and the results are shown below:
>
> | Model | Context Length | Time (s) |
> | :--- | :---: | :---: |
> | TransFourier | 256 | 34,112 |
> | GPT-2 | 256 | 33,839 |
> | Mamba | 256 | 20,006 |
> | Mamba-2 | 256 | 19,390 |
> | TransFourier | 1024 | 34,714 |
> | GPT-2 | 1024 | 40,356 |
> | Mamba | 1024 | 21,075 |
> | Mamba-2 | 1024 | 20,413 |
> | TransFourier | 4096 | 35,863 |
> | GPT-2 | 4096 | 67,290 |
> | Mamba | 4096 | 22,059 |
> | Mamba-2 | 4096 | 21,397 |
> | TransFourier | 8192 | 36,152 |
> | GPT-2 | 8192 | 105,058 |
> | Mamba | 8192 | 22,226 |
> | Mamba-2 | 8192 | 22,129 |
>
> **Summary:**
> As illustrated in the tables, TransFourier demonstrates competitive performance against the updated baselines on all benchmarks. Regarding computational efficiency, while TransFourier does not yet match the highly optimized $O(N)$ CUDA kernels of the Mamba family, its $O(N \log N)$ complexity ensures it significantly outperforms the quadratic $O(N^2)$ Transformer architecture (GPT-2) as sequence length increases. Specifically, at a context length of 8192, TransFourier is nearly $3\times$ faster than GPT-2.
>
> ### **Q3: [Miscellaneous]**
> **Comment:** Some suggestions for paper writing.
>
> **Response:**
>
> Thanks for these helpful suggestions! We will revise our paper accordingly.
>
> ### **Q4: [Other questions]**
> **Comment:** See **questions** section.
>
> **Response:**
>
> 1. We appreciate this valuable suggestion. While we have not yet integrated the FlashFFTConv kernel due to time constraints during the rebuttal period, we fully recognize its potential and we plan to explore this integration in future work to further enhance computational efficiency.
> 2. In fact, FFT-based global token mixing is inherently well-suited for processing sequences with fixed lengths in a non-causal manner. Several prior works exploring spectral methods for bidirectional tasks are discussed in our Related Work section.
> 3. We utilized `torch.float32` for pre-training, as it is currently the required numerical format for performing FFT calculations in PyTorch. We attempted to leverage mixed-precision training using `torch.autocast(device_type=device_type, dtype=torch.bfloat16)` to accelerate the process; however, native `bfloat16` support for FFT operations is not yet available. We believe that as underlying GPU kernels evolve to provide better low-precision support for FFT operations, this architectural paradigm holds significant potential for further efficiency gains and optimization.

---

### Meta-Review · Area_Chair_if87 · 2026-01-09

**Summary:**

The paper proposes a Fourier-based replacement for self-attention with favorable asymptotic complexity and an attention-free design. The approach is conceptually interesting, and reviewers acknowledge the architectural novelty and the effort to provide code and additional experiments.

However, several core concerns remain unresolved under a conservative reading of the record. Key empirical claims—particularly regarding long-context advantages and efficiency—are insufficiently substantiated, with missing or incomplete evidence for inference/generation speed and limited quantitative support for long-context retrieval. Important ablations isolating the contribution of specific architectural components are absent, making it difficult to disentangle architectural effects from training or baseline choices. Comparisons to alternative efficient or linear-attention methods remain incomplete, and some requested validations (e.g., positional encoding variants) were not provided.

Reviewer opinions were mixed and generally cautious, with marginal scores and no explicit post-rebuttal indications of score increases. Given the interrupted review process, this assessment reflects a best-effort, conservative evaluation based solely on the available reviews, rebuttal, and discussion, without assuming any score changes.

**Reviewer Concerns:**

eviewer hUMm:

Addressed:
• Stronger baselines and expanded benchmarks added.
• Causal vs. cross-correlation ablation provided.
• Training-time runtime comparisons added.

Partially addressed:
• Interpretation of reversed subsequence interactions.
• Long-context validation (no clear Needle quantitative results).

Outstanding:
• Ablation of short convolution components.
• Inference/generation speed comparisons.

Reviewer pPta:

Addressed:
• Training runtime experiments added.
• Benchmarks expanded; Llama baseline added.

Partially addressed:
• Motivation for FFT on discrete language.
• Core mixing ablation (limited scope).

Outstanding:
• Clear long-context advantage evidence.
• Generative/decoding behavior and inference speed.
• Robustness and scale concerns.

Reviewer CJE5:

Addressed:
• Training runtime experiments (explicitly acknowledged).
• Modern Transformer baseline added.

Partially addressed:
• Linear-attention comparison and scaling analysis.

Outstanding:
• Explicit positional encoding variants.
• Inference/generation runtime.

**Reviewer Scores:**

Reviewer hUMm:

Original score: 4

Likely post-rebuttal score: 4

Justification:
• Major concerns remain outstanding.

Reviewer pPta:

Original score: 2

Likely post-rebuttal score: 2

Justification:
• No explicit signal of score change; major concerns remain.

Reviewer CJE5:

Original score: 2

Likely post-rebuttal score: 2

Justification:
• Reviewer reiterated unresolved major issues.

---

### Decision · Program_Chairs · 2026-01-26

Reject